# Vitamin D Deficiency Is Significantly Associated with Retinopathy in Type 2 Diabetes Mellitus: A Case-Control Study

**DOI:** 10.3390/nu14010084

**Published:** 2021-12-25

**Authors:** José M. Castillo-Otí, Ana I. Galván-Manso, María R. Callejas-Herrero, Luís A. Vara-González, Fernando Salas-Herrera, Pedro Muñoz-Cacho

**Affiliations:** 1Unidad Vigilancia Epidemiológica e Intervención, 39120 Liencres, Spain; josemaria.castillo@scsalud.es; 2Grupo de Investigación Salud Comunitaria IDIVAL, Primary Care Department, 39007 Santander, Spain; 3Facultad de Enfermería, Universidad de Cantabria, 39008 Santander, Spain; 4Centro de Salud La Marina, 39009 Santander, Spain; anaisabel.galvan@scsalud.es (A.I.G.-M.); rosa.callejas@scsalud.es (M.R.C.-H.); luisvara1@gmail.com (L.A.V.-G.); fresalash@hotmail.com (F.S.-H.); 5Unidad Docente de Medicina Familiar y Comunitaria, 39011 Santander, Spain

**Keywords:** case-control study, diabetes, diabetic retinopathy, screening program, vitamin D deficiency

## Abstract

Aim: Results from meta-analyses point to an association between vitamin D deficiency and the onset of diabetic retinopathy (DR). The objectives of the present study were to evaluate the association of vitamin D for the development of DR and to determine the levels of vitamin D associated with a greater risk of DR. Methods: Between November 2013 and February 2015, we performed a case-control study based on a sample of patients with diabetes in Spain. The study population comprised all patients who had at least one evaluable electroretinogram and recorded levels of 25(OH)D. We collected a series of analytical data: 25(OH)D, 1,25(OH)2D, iPTH, calcium, albumin, and HbA1c. Glycemic control was evaluated on the basis of the mean HbA1c values for the period 2009–2014. A logistic regression analysis was performed to identify the variables associated with DR. Results: The final study sample comprised 385 patients, of which 30 (7.8%) had DR. Significant differences were found between patients with and without DR for age (69.54 vs. 73.43), HbA1c (6.68% vs. 7.29%), years since diagnosis of diabetes (10.9 vs. 14.17), level of 25(OH)D (20.80 vs. 15.50 ng/mL), level of 1,25(OH)2D (35.0 vs. 24.5 pg/mL), treatment with insulin (14.9% vs. 56.7%), hypertension (77.7% vs. 100%), cardiovascular events (33.2% vs. 53.3%), and kidney failure (22.0% vs. 43.3%). In the multivariate analysis, the factors identified as independent risk factors for DR were treatment of diabetes (*p* = 0.001) and 25(OH)D (*p* = 0.025). The high risk of DR in patients receiving insulin (OR 17.01) was also noteworthy. Conclusions: Levels of 25(OH)D and treatment of diabetes were significantly associated with DR after adjusting for other risk factors. Combined levels of 25(OH)D < 16 ng/mL and levels of 1,25(OH)2D < 29 pg/mL are the variables that best predict the risk of having DR with respect to vitamin D deficiency. The risk factor with the strongest association was the treatment of type 2 diabetes mellitus. This was particularly true for patients receiving insulin, who had a greater risk of DR than those receiving insulin analogues. However, further studies are necessary before a causal relationship can be established.

## 1. Introduction

Diabetic retinopathy (DR) is responsible for 2.6% of cases of visual loss throughout the world [1]. In developed countries, it is no longer the main cause (14.4%), having been superseded by hereditary diseases of the retina (20.2%); this is attributed to DR screening programs and improved glycemic control [2,3].

Vitamin D has been associated with different extraskeletal effects, due to the vitamin D receptor (VDR) being present in a wide variety of cells in the body [4,5] and influencing the regulation of more than 2000 genes [6]. While vitamin D deficiency has been associated with multiple diseases, not all of these associations have been verified [7]. However, several meta-analyses found an association between vitamin D deficiency and higher overall mortality [8], lower survival in various cancers [9], or a higher incidence of respiratory diseases [10]. In recent years, several studies examined the association between DR and vitamin D deficiency. The results varied mainly because of the heterogeneous nature of the studies, determined by factors such as population characteristics, vitamin D cutoff points, quality of healthcare, and adjustment for risk factors associated with DR. Two meta-analyses published in 2017 concluded that there was an association between vitamin D deficiency and onset of DR [11,12].

Nevertheless, few studies analyzed this association by controlling for risk factors associated with DR, and results are diverse [13,14,15,16,17,18]. Similarly, levels of risk of DR associated with 25(OH)D and the predictive role of 1,25(OH)2D have not been examined.

The objectives of the present study were to evaluate the association of vitamin D with DR and to determine the levels of vitamin D associated with a greater risk of DR.

## 2. Methods

This was a case-control study based on a sample from an analysis of prevalence of DR in patients with type 2 diabetes mellitus [19] performed between November 2013 and February 2015. The catchment area of the study was the urban area of Santander (northern Spain), which includes 10,744 patients known to have type 2 diabetes mellitus. The target population comprised patients from the study area diagnosed with type 2 diabetes mellitus according to their clinical history. Multistage cluster random sampling was performed. In the first stage, we selected one of the nine health centers from the metropolitan area (La Marina Health Center). In the second stage, we selected groups of patients from the center until the sample size was reached. Both stages were performed using simple random sampling, with probability proportional to the size of the population. Physicians updated the lists of patients with type 2 diabetes mellitus in order to avoid errors in diagnosis.

We selected all patients included in the prevalence study who had at least one evaluable retinal scan and recorded levels of 25(OH)D. We defined a case as a patient with any degree of DR and a control as a patient who did not have DR. The main endpoint was 25(OH)D level. We also analyzed other covariates identified as possible risk factors for DR, namely, age, years since the diagnosis of type 2 diabetes mellitus, level of glycemic control, treatment of type 2 diabetes mellitus, control of arterial blood pressure, lipid control, obesity, smoking, 1,25(OH)2D levels, chronic kidney disease, and cardiovascular events [20].

The prevalence study identified 30 patients with DR and 355 without DR. The 25(OH)D levels were available for all patients. Therefore, for an alpha risk of 5% and a beta risk of 20% in a two-tailed test and assuming a 40% prevalence of 25(OH)D deficiency [21], the study could detect odds ratios ≥3.

The patients included in the study were informed by letter and contacted via telephone by the Ophthalmology Clinic of Hospital Universitario Marqués de Valdecilla, where they underwent testing of visual acuity using Snellen optotypes with the patient’s graduation and nonmydriatic retinography (Zeiss Visucam PRO NM) to obtain two photographs at 45° (one centered on the macula and the other on the optic disc following the EURODIAB protocol) [22]. The photographs were taken by a single professional with experience comprising >600 previous scans and evaluated by a retina specialist according to the categories of the International Clinical Diabetic Retinopathy Severity Scale [23].

We performed an analytical workup in the Endocrinology Laboratory of Hospital Universitario Marqués de Valdecilla. This included 25(OH)D, 1,25(OH)2D, intact parathyroid hormone (iPTH), calcium, albumin, and HbA1c. Whole-blood HbA1c levels were determined using high-pressure liquid chromatography. Levels of iPTH and 25(OH)D were evaluated using a specific automated chemiluminescent immunoassay. Levels of 1,25(OH)2D were analyzed using a radioimmunoassay.

Patient data were obtained from the electronic clinical history of the Servicio Cántabro de Salud (Cantabrian Health Service). The clinical variables analyzed were age, sex, weight, height, body mass index (BMI), years since diagnosis of type 2 diabetes mellitus, current treatment of type 2 diabetes mellitus, family history of type 2 diabetes mellitus, degree of control of arterial blood pressure and dyslipidemia, cardiovascular events, chronic kidney disease, calcium/vitamin D supplements, pregnancy, and smoking.

Glycemic control was evaluated on the basis of the mean HbA1c values for the period 2009–2014. Lipid control was evaluated on the basis of the mean values of total cholesterol, HDL-cholesterol (HDL-c), LDL-cholesterol (LDL-c), and triglycerides during the same period. The degree of glycemic and lipid control was analyzed following the 2015 recommendations of the American Diabetes Association [24]. High blood pressure (HBP) was evaluated by recording the mean pressures for the same period following the recommendations of the Eighth Joint National Committee (JNC8) for patients with type 2 diabetes mellitus [25]. Good control of blood pressure was defined as mean values lower than 140/90 mmHg or 130/80 mmHg if chronic kidney disease (any stage) was recorded. Grade I hypertension was defined as 140–159 mmHg (systolic) and 90–99 mmHg (diastolic). Grade II hypertension was defined as 160–179 (systolic) and 100–109 (diastolic). Kidney failure was evaluated using the classification criteria of Kidney Disease Improving Global Outcomes [26].

The procedures were carried out after signature of the informed consent document, which was approved by the Clinical Research Ethics Committee of Cantabria (Acta 4/2014). Data were anonymized and treated confidentially according to Law 41/2002 (dated 14 November) and Law 7/2002 (dated 10 December) for Healthcare Standards of Cantabria.

The statistical analysis was performed using IBM SPSS Statistics Version 25.0 (IBM Corp, Armonk, NY, USA) and MedCalc Statistical Software Version 18.9 (MedCalc Software bvba, Ostend, Belgium; http://www.medcalc.org; 2018) (accessed on 12 December 2021). The main qualitative variables were reported using percentages with 95% confidence intervals (CIs). The normality of the distribution of quantitative variables was assessed using the Kolmogorov–Smirnov test. Variables were expressed, when applicable, as the mean (SD) or median (IQR). The DeLong procedure (1988) was used to compare the area under the receiver operating characteristic (ROC) curve.

Hypothesis testing (univariate analysis) was performed using the chi-squared test or the Fisher exact test for qualitative variables. The *t*-test or Mann–Whitney test was used for quantitative variables. Statistical significance was set at *p* < 0.05.

In order to identify the variables associated with DR, we performed a logistic regression analysis. The dependent variable was dichotomized as presence or absence of DR. The latter category included any grade of DR. The Youden index was used to identify levels of 25(OH)D and 1.25(OH)2D associated with prediction of DR. A univariate analysis was performed by estimating the OR and significance of each independent variable with the Wald statistic. Variables in the multivariate analysis were selected using the Hosmer–Lemeshow test, as described elsewhere [27] and according to which the variables initially included in the univariate analysis were those with a level of significance of *p* < 0.25. The backward elimination (conditional) method was used for automatic selection of variables in the final model. This model was compared with others with the same number of variables according to the pathophysiology and current knowledge of the subject. The ability of the different models to predict DR was compared using the area under the curve (AUC) [28]. The criterion followed was that of including one independent variable for every 10 cases of the dependent variable in the final model [29].

## 3. Results

### 3.1. Characteristics of the Study Sample

Of the 497 patients who fulfilled the inclusion criteria, 16 could not be traced, 24 refused to participate, five were eliminated owing to physical disability, 12 were eliminated because they did not attend the visit on two occasions, and 55 did not have recorded levels of 25(OH)D. The final sample comprised 385 patients (Figure 1). Table 1 shows the clinical and epidemiological characteristics of patients with and without DR. Significant differences were found for age (69.54 vs. 73.43 years), HbA1c level (6.68% vs. 7.29%), years since diagnosis of type 2 diabetes mellitus (10.9 vs. 14.17), level of 25(OH)D (20.80 vs. 15.50 ng/mL), level of 1,25(OH)2D (35.0 vs. 24.5 pg/mL), patients receiving insulin (14.9% vs. 56.7%), hypertensive patients (77.7% vs. 100%), cardiovascular events (33.2% vs. 53.3%), and kidney failure (22.0% vs. 43.3%).

As for treatment of type 2 diabetes mellitus, 12.9% of patients were treated with diet, 69.0% were treated with oral antidiabetic drugs (OAD), 11.4% were treated with OAD + insulin, and 6.7% were treated with insulin. The most commonly used drug was metformin (68.2%).

Of the 385 patients for whom evaluable retinography data and 25(OH)D levels were available, 30 had DR (any grade; 7.80, 95% CI: 5.32, 10.94).

### 3.2. Variables Associated with DR

Mean HbA1c during the study period was 6.87% (0.96) (median, 6.71%); 67.0% of patients had a mean of <7% and 89.4% had HbA1c <8%. HBP was recorded in 79.5% of patients, with a mean blood pressure of 140/76.5 mmHg and a mean of 12.8 years since diagnosis. The prevalence of dyslipidemia was 66.2%. The degree of control for each lipid factor was as follows: LDL-c = 25.5%; HDL-c = 66.6%; triglycerides = 64.9%. Good control for all three variables was observed in only 13.7% of patients.

Values associated with the Youden index for 25(OH)D and 1,25(OH)2D were ≤16 ng/mL and ≤29 pg/mL, respectively.

In order to analyze risk factors for DR, we performed a univariate analysis for each factor (Table 2). The results were not significant for dyslipidemia or smoking. Therefore, the variables selected for the multivariate analysis were age, years since diagnosis, BMI, glycemic control, treatment of type 2 diabetes mellitus, arterial hypertension, cardiovascular events, chronic kidney disease, 25(OH)D, and 1,25(OH)2D.

We created a composite variable—combined vitamin D—by combining levels of 25(OH)D and 1,25(OH)2D according to the Youden indexes and observed that patients with both metabolites below these levels had a greater risk of DR (OR 5.21, 95% CI: 1.76, 15.42; *p* = 0.003) (Table 3).

In the initial multivariate model, with the 10 variables selected from the univariate analysis, we used the backward elimination (conditional) method to construct Model 1, which included treatment of type 2 diabetes mellitus, degree of control of HBP, and 25(OH)D (Appendix A). The significantly associated variables were treatment of diabetes (*p* = 0.001) and 25(OH)D (*p* = 0.025). The high risk of DR in patients receiving insulin (OR 17.01) was also noteworthy. The AUC for this model was 78%.

We also proposed two further models. In Model 2, we used 1,25(OH)2D, and, in Model 3, we used the composite variable obtained from the level of control of 25(OH)D and 1,25(OH)2D (Table 4). The three models yielded similar AUCs, around 76–79%, with no significant differences (Appendix A).

The significantly associated variables (*p* < 0.05) in the three models were treatment of diabetes and the different variables associated with vitamin D.

In the third model, in the case of patients with low levels of both metabolites, the OR was 4.48 (95 CI%: 1.43, 13.99) (Table 4). The AUC for this model was 79.3%.

Collinearity was analyzed using a tolerance test, which yielded a negative result for all of the variables included in the multivariate models.

## 4. Discussion

Vitamin D was measured throughout the year, albeit only once per patient. No seasonal corrective factors were applied. While this could prove to be a limiting factor for the study results, values were distributed similarly over all four seasons. Another possible limitation of the study is that some variables such as sun exposure, diet, or socioeconomic factors were not taken into account.

The crude OR for 25(OH)D deficiency (<20 ng/mL) was 2.31 (95% CI: 1.02, 5.22; *p* = 0.044), which was higher than that recorded in the meta-analysis of Luo et al. [12] (OR 2.03) and Zhang et al. [11]. (OR 1.27). A similar result was obtained in a study performed in China (OR 1.93) [30]. In our meta-analysis [13,31,32,33,34,35,36,37,38,39], which included studies up to 2016 (Figure 2), those studies with a cutoff point of 20 ng/mL had an OR of 1.51 (95% CI: 1.16, 1.97). Therefore, 25(OH)D deficiency was associated with DR. The OR adjusted for risk factors remained at 2.47 (95% CI: 1.07, 5.52) in Model 1.

It is important to highlight that, in the multivariate analysis, treatment of diabetes was the risk factor that was most strongly associated with DR, especially in patients receiving insulin. This association has been observed in other national studies with high ORs [40,41,42,43], as well as in international studies [44,45,46,47]. A cohort study [48] found that DR was more frequent during follow-up in patients who received insulin (11.3%) than in patients who continued to take oral hypoglycemic agents (6.8%); however, the authors did not provide an explanation for this finding. Similarly, no explanations were provided for the mechanism underlying this association, which remained unchanged after adjustment for other risk factors and, in the case of cardiovascular events, was not observed (*p* = 0.86). We applied a variable whose categories were OAD or diet (reference category), human insulin, or insulin analogue. When we compared the presence or absence of DR with this variable, we obtained a crude OR of 12.39 (95% CI: 4.46, 34.43; *p* < 0.001) for human insulin (Table 5). In the multivariate analysis, after controlling for the variables age, HbA1c, years since diagnosis, hypertension, and both metabolites of vitamin D, the OR was 5.76 (95% CI: 1.16, 28.69; *p* = 0.033) for human insulin (Appendix A). Therefore, treatment with human insulin carries a greater risk than treatment with insulin analogues and is significantly associated with DR. Human insulin per se may cause an inflammatory or immune reaction. Excipients or retardants may also be responsible for this association. Other possible causes are fluctuations in blood sugar caused by the rapid action of human insulin.

While 16 ng/mL is the cutoff point for 25(OH)D that best predicts the risk of DR (OR 2.43), patients with levels <20 ng/mL had a similar risk (OR 2.29), as shown in Appendix A. The behavior of the combined variable (vitamin D metabolites) was particularly interesting, since, when both metabolites were below their Youden index, the risk of DR multiplied (OR 5.21, 95% CI: 1.76, 15.42; *p* = 0.003). This effect was maintained, as can be seen in Model 3 of the multivariate analysis. Therefore, we can consider it a good tool for predicting the onset of DR. This effect may be due to the fact that low levels of both metabolites were observed in patients with severe vitamin D deficiency maintained over time. Therefore, we believe that the measurement of both metabolites is justified, since 1,25(OH)2D is useful for identifying those patients who are at the highest risk.

## 5. Conclusions

In conclusion, the levels of 25(OH)D and treatment of diabetes are significantly associated with DR after adjusting for other risk factors.

Combined levels of 25(OH)D < 16 ng/mL and 1,25(OH)2D < 29 pg/mL are the best predictors of the risk of having DR, with respect to vitamin D deficiency.

While not the objective of our study, the risk variable with the strongest association was treatment of type 2 diabetes mellitus. This was particularly true for patients receiving insulin, who had a greater risk of DR than those receiving insulin analogues. However, further studies are necessary before a causal relationship can be established.

## Figures and Tables

**Figure 1 nutrients-14-00084-f001:**
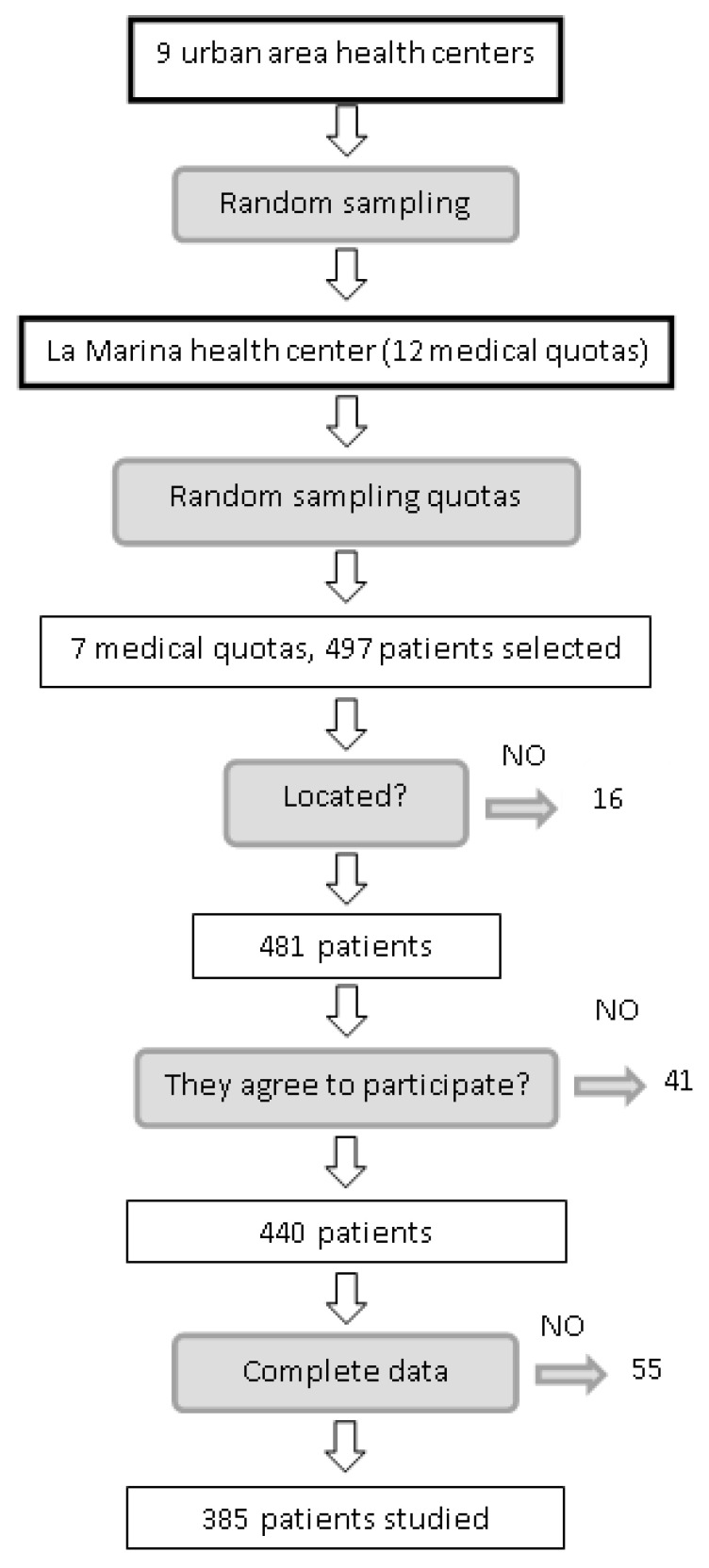
Selection process for patients included in the sample.

**Figure 2 nutrients-14-00084-f002:**
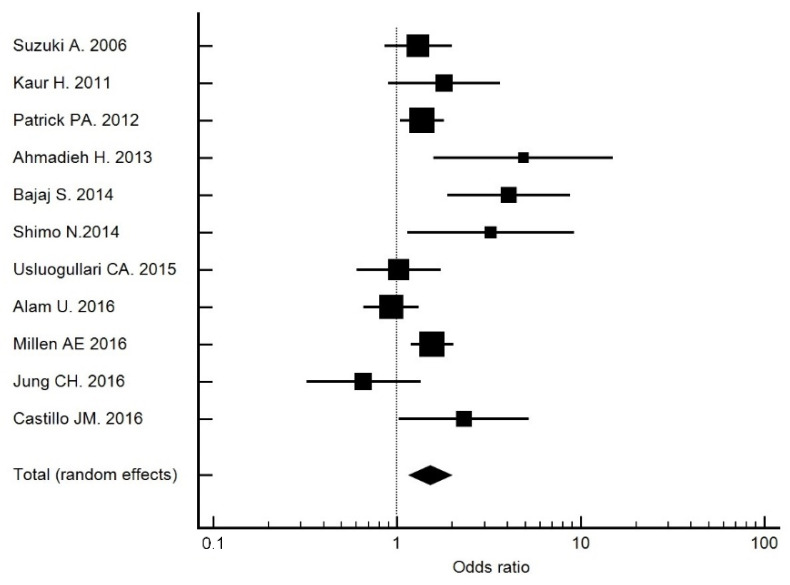
Forest plot of studies with a cutoff point of 20 ng/mL. The total OR was 1.51 (95% CI: 1.16, 1.97).

**Table 1 nutrients-14-00084-t001:** Clinical and epidemiological characteristics.

Variable	Total	No DR	DR	
	No. (%) or mean (SD)	No. (%) or mean (SD)	No. (%) or mean (SD)	*p*-Value
Age	69.89 (9.86)	69.54 (9.95)	73.43 (8.08)	0.040
Sex				0.592
Men	213 (55.30)	195 (54.90)	18 (60)	
Women	172 (44.70)	160 (45.10)	12 (40)	
Years since diagnosis	10.99 (7.06)	10.69 (6.89)	14.57 (8.11)	0.016
Body mass index	29.90 (4.77)	29.98 (4.85)	28.94 (3.76)	0.164
HbA1c ^a^	6.71 (6.17, 7.40)	6.68 (6.13, 7.33)	7.29 (6.68, 8.08)	0.001
25(OH)D ^a^	20 (13.3, 28)	20.80 (14, 28)	15.50 (10.8, 23.3)	0.013
<10 ng/mL	28 (7.30)	22 (6.20)	6(20)	
10–31.99	289 (75.10)	267 (75.20)	22 (73.30)	
≥32 ng/mL	68 (17.70)	66 (18.60)	2 (6.70)	
1,25(OH)2D pg/mL ^a^	33 (20–53)	35 (21, 54)	24.50 (13, 38.25)	0.021
Patients on insulin	70 (18.20)	53 (14.90)	17 (56.70)	<0.001
HBP	306 (79.50)	276(77.70)	30 (100)	0.004
Dyslipidemia	255 (66.20)	239 (67.30)	16 (53.30)	0.120
Smoking	58 (16.34)	54 (15.20)	4 (13.30)	0.838
Cardiovascular events	134 (34.80)	118 (33.20)	16 (53.30)	0.027
Kidney failure	91 (23.60)	78 (22)	13 (43.30)	0.008
Diabetic retinopathy	30 (7.80)			

DR: Diabetic Retinopathy. HBP: High blood pressure. ^a^ Nonparametric variables. Expressed as median (interquartile range).

**Table 2 nutrients-14-00084-t002:** Univariate analysis of risk factors.

	OR	95% CI	*p*-Value
Female sex	0.813	0.380, 1.737	0.592
Age (years)	1.044	1.002, 1.089	0.041
Age at diagnosis	1.000	0.964, 1.036	0.987
Years since diagnosis	1.062	1.018, 1.107	0.005
Body mass index	0.952	0.877, 1.034	0.237
Normal weight (reference value)			0.230
Obesity G-1	1.330	0.426, 4.152	0.624
Obesity G-2	0.639	0.188, 2.169	0.473
Glycemic control			
HbA1c(<7) (reference)			0.005
HbA1c (7–7.9)	2.911	1.216, 6.968	0.016
HbA1c (8–10)	5.148	1.827, 14.511	0.002
HbA1c (>10)	11.400	0.952, 136.482	0.055
Treatment of diabetes			
Diet (reference value)			<0.001
OAD	2.324	0.295, 18.286	0.423
Insulin	21.778	2.542, 186.566	0.005
OAD + insulin	12.600	1.526, 104.035	0.019
HBP (yes)	4.984	1.175, 21.130	0.029
Cardiovascular events (yes)	2.295	1.084, 4.862	0.030
No events (reference value)			0.073
1 event	1.929	0.804, 4.628	0.141
2 events	3.276	1.173, 9.150	0.024
3 events	--	--	--
4 events	16.929	1.005, 285.073	0.050
Dyslipidemia			
Good control (reference value)			0.802
Poor control 1 factor	0.633	0.245, 1.635	0.345
Poor control 2 factor	0.688	0.248, 1.909	0.473
Poor control 3 factor	0.619	0.185, 2.068	0.436
Kidney disease			
Normal (reference value)			0.109
Grade 2 (mild)	1.849	0.589, 5.807	0.293
Grade 3a	2.696	0.840, 8.655	0.096
Grade 3b	4.044	1.218, 13.429	0.022
Grade 4	8.088	0.698, 93.722	0.094
Grade 5	--	--	--
Smoking	0.892	0.299, 2.665	0.838
25(OH)D	0.947	0.906, 0.991	0.018
25(OH)D Categorized			
>16 ng/mL (reference value)			0.001
≤16 ng/mL	2.427	1.142, 5.160	0.021
1,25(OH)2D			
>29 pg/mL (reference value)			0.001
≤29 pg/mL	3.313	1.338, 8.205	0.010

OAD: Oral antidiabetic drugs; HBP: High blood pressure.

**Table 3 nutrients-14-00084-t003:** Risk according to levels of 25(OH)D combined with 1,25(OH)2D.

Variable	*p*-Value	OR	95% CI
25(OH)D > 16 ng/mL and 1,25(OH)2D > 29 pg/mL	0.011		
25(OH)D > 16 ^a^ and 1,25(OH)2D ≤ 29 ^b^	0.882	1.14	0.21, 6.04
25(OH)D ≤ 16 ^a^ and 1,25(OH)2D > 29 ^b^	0.341	1.85	0.52, 6.61
25(OH)D ≤ 16 ^a^ and 1,25(OH)2D ≤ 29	0.003	5.21	1.76, 15.42
Constant	<0.001	0.04	

^a^ ng/mL; ^b^ pg/mL.

**Table 4 nutrients-14-00084-t004:** Multivariate analysis models.

	*p*-Value	OR	95% CI	AUC (%)
Model 1				76.3
Treatment of diabetes (reference: Diet)	0.000			
Oral antidiabetic drugs (OAD)	0.436	2.28	0.29, 18.11	
Insulin	0.004	24.93	2.80, 221.65	
OAD + insulin	0.028	10.95	1.30, 92.32	
HBP (reference: Normal BP)	0.167			
Stage 1	0.073	2.28	0.93, 5.62	
Stage 2	0.229	2.62	0.55, 12.58	
Levels of 25(OH)D (≤16 ng/mL)	0.027	2.47	1.11, 5.52	
Model 2				78.2
Treatment of diabetes (reference: Diet)	0001			
Oral antidiabetic drugs (OAD)	0.660	1.60	0.20, 13.02	
Insulin	0.035	11.65	1.18, 114.64	
OAD + insulin	0.055	8.23	0.96, 70.78	
HBP (reference: Normal BP)	0.150			
Stage 1	0.068	2.57	0.93, 7.10	
Stage 2	0.172	3.22	0.60, 17.26	
Levels of 1,25(OH)_2_D (<29 pg/mL)	0.038	2.73	1.06, 7.07	
Model 3				79.3
Treatment of diabetes (reference: Diet)	0.001			
Oral antidiabetic drugs (OAD)	0.523	1.99	0.24, 16.54	
Insulin	0.022	15.15	1.49, 154.15	
OAD + Insulin	0.041	9.67	1.10, 85.10	
HBP (reference: Normal BP)	0.145			
Stage 1	0.064	2.66	0.94, 7.48	
Stage 2	0.181	3.26	0.58, 18.47	
25(OH)D > 16 ng/mL and 1,25(OH)2D > 29 pg/mL	0.046			
25(OH)D > 16 ^a^ and 1,25(OH)2D ≤ 29 ^b^	0.507	1.57	0.41, 5.99	
25(OH)D ≤ 16 ^a^ and 1,25(OH)2D > 29 ^b^	0.706	1.39	0.25, 7.77	
25(OH)D ≤ 16 ^a^ and 1,25(OH)2D ≤ 29 ^b^	0.010	4.48	1.43, 13.99	

^a^ ng/mL; ^b^ pg/mL; AUC: Area under the curve; HBP: High blood pressure.

**Table 5 nutrients-14-00084-t005:** Risk of DR according to type of insulin.

	*p*-Value	OR	95% CI
Diet or OAD (reference)	<0.001		
Insulin analogues	<0.001	5.65	2.26, 14.12
Human insulin	<0.001	12.39	4.46, 34.43

OAD: Oral antidiabetic drugs.

## Data Availability

The data presented in this study are available on request from the corresponding author.

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
