# Peer review of "Vitamin D Deficiency Is Significantly Associated with Retinopathy in Type 2 Diabetes Mellitus: A Case-Control Study"

_nutrients, 2021, doi:10.3390/nu14010084_

Round 1

Reviewer 1 Report

Many thanks for your contribution.

You showed many comparison data, however, as you described, 'an independent risk factor'

is not a suitble one. Only there is an association between low D levels and presence of DR, not develepment

of DR.

By your case-control study, it is hard to say vitamin D is an independent risk factor, rather you had better

describe low D levels have an association with the presence of DR....

Author Response

We agree with comments and have changed the title of the article:

            Vitamin D deficiency is significantly associated with retinopathy in type 2 diabetes mellitus. A case control study.

In addition to:

Replace “independent risk factors” with “significantly associated with” in the results section (twice) and in the discussion section (twice) and in the conclusions.

Reviewer 2 Report

Dear Authors,

This work may be interesting, but the form of sent text is terrible.

  1. Very low quality of english.
  2. Title not adequete to content
  3. Very poor quality of introduction-  too short, described studies without references.
  4. a lot of mental shortcuts in all text e.g. "mean hemoglobin"
  5. In one place is written 385 DR patients, in other place 30 DR patients..so what's the number od patients???
  6. Whole article is very chaotic
  7. In my opinion conclusions should be as a separate point

Author Response

  1. We request the revision of English by MAPI.
  2. We have changed the title:

            Vitamin D deficiency is significantly associated with retinopathy in type 2 diabetes mellitus. A case control study.

  1. A paragraph has been added in the introduction; the references of the studies described are clarified.
  2. “Mean hemoglobin” is corrected by “mean HbA1c”.
  3. The total number of patients and those with diabetic retinopathy is clarified.
  4. No coment
  5. A separate point is made for conclusions.

We agree with the comments and suggestions proposed.

Reviewer 3 Report

Very interesting project on association of vitamin D and diabetic retinopathy.

A body of literature exists on vitamin D and other factors of health. Would be helpful to summarize some interesting findings about vitamin D in other features of diabetes and/or ophthalmic health, and to postulate why this relationship may be seen. Worth discussion of exercise (and sunlight exposure), diet (dietary intake of vit D), and socioeconomics as these factors are not controlled in the presented models but likely influence the diabetes disease state.

In abstract line 40 and in conclusion lines 246-247, the authors recommend supplementation with vitamin D. Is this based on existing health recommendations or the authors’ research here? If existing recommendations, citation should be made. If the authors’ research, this recommendation is too strong from the current research. There is no evidence from this manuscript that supplementation with vitamin D improves diabetic retinopathy, as the study was not designed in this fashion. Further research should be recommended, as in the final concluding sentence.

Figures do not appear to be available in reviewer manuscript or supplemental file.

Author Response

  1. We have added a paragraph in the introduction commenting on the effect of vitamin D deficiency on respiratory infections, total mortality and cancer.

  1. have been added as limitations that have not been controlled by: exercise, diet and socioeconomic factors, factors that can influence the complications of diabetes.

  1. The recommendation to supplement with Vitamin D has been deleted in the summary and in the conclusions.

  1. We attach figures 1 and 2.
